# A Systematic Review on Human Thermal Comfort and Methodologies for Evaluating Urban Morphology in Outdoor Spaces

Iago Turba Costa [1], Cassio Arthur Wollmann [1,*], Luana Writzl [1], Amanda Comassetto Iensse [1], Aline Nunes da Silva [1], Otavio de Freitas Baumhardt [1], João Paulo Assis Gobo [2], Salman Shooshtarian [3] and Andreas Matzarakis [4,5]

[1] Department of Geosciences, Natural and Exact Sciences Center, Federal University of Santa Maria, Santa Maria 97105-900, Brazil; iago.costa@acad.ufsm.br (I.T.C.); luana.writzl@acad.ufsm.br (L.W.); amanda.comassetto@acad.ufsm.br (A.C.I.); silva.aline@acad.ufsm.br (A.N.d.S.); otavio.baumhardt@acad.ufsm.br (O.d.F.B.)

[2] Department of Geography, Core of Exact Earth Sciences, Federal University of Rondônia, Porto Velho 76801-059, Brazil; joao.gobo@unir.br

[3] School of Property, Construction and Projetc Management, Royal Melbourne Institute of Technology (RMIT), Melbourne, VIC 3001, Australia; salman.shooshtarian@rmit.edu.au

[4] Institute of Earth and Environmental Sciences, University of Freiburg, D-79104 Freiburg, Germany; andreas.matzarakis@meteo.uni-freiburg.de

[5] Democritus University of Thrace, 69100 Komotini, Greece

\* Correspondence: cassio@ufsm.br

**Abstract:** The exponential growth of urban populations and city infrastructure globally presents distinct patterns, impacting climate change forecasts and urban climates. This study conducts a systematic review of the literature focusing on human thermal comfort (HTC) in outdoor urban environments. The findings indicate a significant surge in studies exploring HTC in open urban spaces in recent decades. While historically centered on Northern Hemisphere cities, there is a recent shift, with discussions extending to various metropolitan contexts in the Southern Hemisphere. Commonly employed urban categorization systems include Sky View Factor (SVF), Height × Width (H/W) ratio, and the emerging Local Climate Zones (LCZs), facilitating the characterization of urban areas and their usage. Various thermal indices, like Physiological Equivalent Temperature (PET), Predicted Mean Vote (PMV), Universal Thermal Climate Index (UTCI), and Standard Effective Temperature (SET), are frequently utilized in evaluating external HTC in metropolitan areas. These indices have undergone validation in the literature, establishing their reliability and applicability.

**Keywords:** systematic review; human thermal comfort; verticalization; urban canyon; human thermal comfort; outdoor thermal comfort

## 1. Introduction

Considering the growth of cities and the various microclimates generated by the diversity of morphologies in the urban environment, several studies conducted around the world have attempted to understand the relationship between external human thermal (dis)comfort and the various environments found within the urban environment [1–5].

According to urban morphology [6–8], people who move along, around, and outside the urban areas of a city are increasingly exposed to a variety of conditions that vary in air temperature, humidity, wind, and sunshine (or lack thereof). Such morphological differences result in different microclimates, which may make the environment (un)comfortable for the population that transits through it [2,9], and it is well known that extreme environmental variables can cause cardiovascular diseases, as previously

demonstrated in research [10–14], in addition to interfering with basic day-to-day external activities [10,13,15,16].

External areas must be of good quality and livable in order for people to carry out their daily activities in greater comfort [3]. Thus, a concern for a city's and its inhabitants' health arises, as represented by HTC in external environments, which has been the subject of countless studies in recent decades, primarily in cities with a high degree of urbanization and verticalization, as in the works of [10–13].

In the urban setting of cities with a high degree of urbanization, there is a concentration of buildings in their landscape, particularly in the cities' most central districts, where buildings tend to have several stories, resulting in a phenomenon known as urban canyon [14]: "The urban canyon has been used in urban climatology as a main concept for describing the basic pattern of urban space defined by two adjacent buildings and the ground plane", write [14].

Verticalization is a notion in the urban environment that is associated with terminologies such as buildings, skyscrapers, and buildings [14,15]. Verticalization, on the other hand, is also a process of city consolidation, as it provides the circumstances and amplified changes of vertical development associated with metropolitan regions and urban surroundings of already consolidated major cities [15,16].

Several studies have attempted to integrate personal sensory features with environmental variables in order to establish ambient settings that may be advantageous to the maximum number of individuals, with the goal of finding solutions to reduce the impacts and enhance future outdoor urban environments [17]. To that purpose, numerous analytic approaches may be used to externally assess meteorological data and describe local urban morphology [17,18].

Several literature studies were conducted in this regard, with an emphasis on exterior thermal (dis)comfort in the metropolitan environment across the world. The authors of [19] intended to explore studies on outdoor thermal comfort in the Australian setting, with the goal of learning about the topic of research, applicable methodology, data-gathering methods, and findings. The authors of [20] investigated the probable impacts of green infrastructures in urban areas on thermal comfort, whereas [21] sought investigations on the urban microclimate and exterior thermal comfort in public places, with an emphasis on hot–humid cities.

When performing massive bibliographic searches, systematic reviews are critical for making scientific and cohesive discoveries of existing publications on the same subject [22,23]. The systematic review varies from the standard review in that it tries to conduct scientific research. It guides the researcher through a series of key procedures that may vary depending on the study author's definition [22,24]. It helps the researcher to create important criteria and objectives for gathering data that will contribute to the solution of the research problem. The authors of [22] add that a systematic review supports a huge amount of gathered studies, broadening the search and structuring the data to meet the researcher's goals.

This research conducted a systematic review to answer the question: "What are the analysis methods present in research that involve the relationship between urban environments with external human thermal (dis)comfort, at the level of pedestrians?".

## 2. Materials and Methods

A systematic review is a research tool that allows you to conduct a bibliographic review on a given topic. The systematic approach is being used increasingly frequently in health sciences, allowing for the collection and summarization of existing data, as well as the reaffirmation of research hypotheses and the definition and knowledge of research techniques [25]. An SR varies from a standard review, often known as a narrative review, in that it answers more specific questions and research concerns [26].

Having access to texts and research that deal directly with the researched object is one of the main benefits of conducting a systematic review; this allows for a more efficient

discussion of the concepts worked. According to [27], narrative reviews frequently employ informal approaches, which might alter how selected research is acquired and processed, implying that the researcher's own beliefs can be reinforced within the process.

The systematic review employs quantitative or qualitative methods and practices [28]. The authors also suggest that a review can be more impactful when conducted in a group (of two or three academics), allowing for discussion of the topic of interest and sharing of the findings.

Five phases are required for systematic review construction [29]: (a) a question definition, which will serve as the beginning premise for the SR and must be clear and objective; (b) looking for evidence utilizing the research topic's keywords, using electronic database search engines and sources that can answer the question to be examined; this search must be for the keywords; (c) study selection, which should take into consideration titles, abstracts, and total reading time, as well as pre-defined inclusion and exclusion criteria; (d) methods analysis, including a thorough reading of the publications to ensure that the required criteria are satisfied; and (e) presentation and discussion of results, which must be highly comprehensive in order to be reproduced in the future.

This study's systematic review included seven adopted stages [30,31]: 1—Elaboration of the research question; 2—Determination of search terms; 3—Database selection and bibliography search; 4—Work selection and exclusion; 5—Complete reading of the literature chosen; 6—Results presentation; 7—Definition of the study technique. According to the review results, the major goal of conducting this research's systematic review is connected to stage (7), which aided in the design of the research technique.

At the base of the research is the initial question for its execution; in this sense, the same question presented with the justification of this research was formulated and used, namely, "What are the analysis methods present in research that involve the relationship between urban environments with external human thermal (dis)comfort, at the level of pedestrians?".

Keywords are identified as a technique to check similarities between previously created investigations and whether they are relevant to the research and therefore may be included to assist in answering the query. In this sense, words should be used that express the objectives that one wishes to find in the articles. The keywords to answer the research question were "Verticalization" OR "Urban Canyon" AND "Human thermal comfort" AND "Pedestrian" AND "outdoor thermal comfort" AND "Urban exterior".

To carry out the search for relevant bibliographies, the translated keywords and online platform search engines were used; the following platforms were used:

(A). Scopus (https://www.scopus.com/home.uri, accessed on 13 February 2024);
(B). Connected Papers (https://www.connectedpapers.com, accessed on 13 February 2024);
(C). Science Direct (https://www.sciencedirect.com, accessed on 13 February 2024).

Movements of inclusion and exclusion of searches were carried out in the selection procedures, with three different rounds of selection carried out: (A) reading of titles and keywords; (B) reading of all summaries (abstracts); and (C) full reading of all articles that were selected for analysis assembly.

Papers with titles that implied a link with HTC, verticalization, urban canyons, outdoor spaces, or the influence of urban canyons were evaluated for title selection. Those that did not display object and focus in the aforementioned criteria were disqualified.

All abstracts were read in the second round of selection and, if selected, they were separated for the entire reading of the paper and linking the urban process to verticalization and/or urban canyons. The abstracts that were rejected for the full-article study were from studies that exclusively concentrated on computational modeling, without an emphasis on human evaluation, only on meteorological factors, and just on wind speed and its modeling. The acceptable publications were the articles in which the whole reading was completed, and the findings comprised the final phase, namely the meta-analysis.

In the meta-analysis, the assessment criteria are determined by reading the evaluators; in this sense, the characteristics examined were (A) the research nation; (B) the city where the study was conducted; (C) the location factor, whether continental or coastal; (D) the year of publication; (E) the climatic classification of the research region; (F) the season of the year in which the survey was conducted; (G) data collecting time; (I) thermal comfort indices used; (K) the canyon's defining parameter; (L) data collecting method; (M) meteorological data gathering instruments; (N) tabulation software; (O) interviewing; and (P) total number of interviews gathered [25].

## 3. Results

### 3.1. Meta-Analysis

A total of 988 items were discovered when conducting searches on the three web platforms using the descriptors specified in the technique. Scopus had 36 articles, Connected Papers had 196 papers, and Science Direct had 756 publications. The examination of the characteristics set for the development of the systematic review, as well as the exclusion of publications that did not match with the intended aims of measuring human thermal comfort in open urban contexts, was thus begun. Table 1 indicates how many articles were accepted and rejected at each level.

**Table 1.** Papers selected and rejected at each stage of the systematic review on each scientific research platform.

| Research Platform | Total Articles in the First Search | Selected Titles | Rejected Titles |
|---|---|---|---|
| Scopus | 36 | 16 | 20 |
| Connected Papers | 196 | 82 | 114 |
| Science Direct | 756 | 275 | 481 |
| Total | 988 | 373 | 615 |
| **Abstracts evaluated in each platform** | **Total Abstract evaluated** | **Selected Abstracts** | **Rejected Abstracts** |
| Scopus | 16 | 14 | 2 |
| Connected Papers | 75 | 41 | 34 |
| Science Direct | 275 | 63 | 212 |
| Total | 364 | 118 | 248 |
| **Articles selected for full reading** | **Articles selected for full reading** | **Articles selected for full reading (%)** | **Articles selected for full reading (%)** |
| Scopus | 14 | 14.0 | 14.0 |
| Connected Papers | 36 | 36.0 | 36.0 |
| Science Direct | 50 | 50.0 | 50.0 |
| Total | 100 | 100.0 | 100.0 |

The platforms Scopus, Connected Papers, and Science Direct were used in that order. As a result, before the complete reading stage was reached, some articles were found to be repeated across platforms. As a result, five articles from Connected Papers were excluded because they were already included in Scopus, and eighteen articles from Science Direct were also excluded because they were already included in other platforms.

There has been a significant rise in the number of papers that research HTC in cities with high vertical expansion in the last decade, particularly after 2013. More than ten publications were published in the years 2020, 2021, and 2023, indicating that there is a strengthening focus on HTC in urban contexts.

Figure 1 shows that 100 papers were thoroughly examined, with the earliest from 1997 and the most current from 2023.

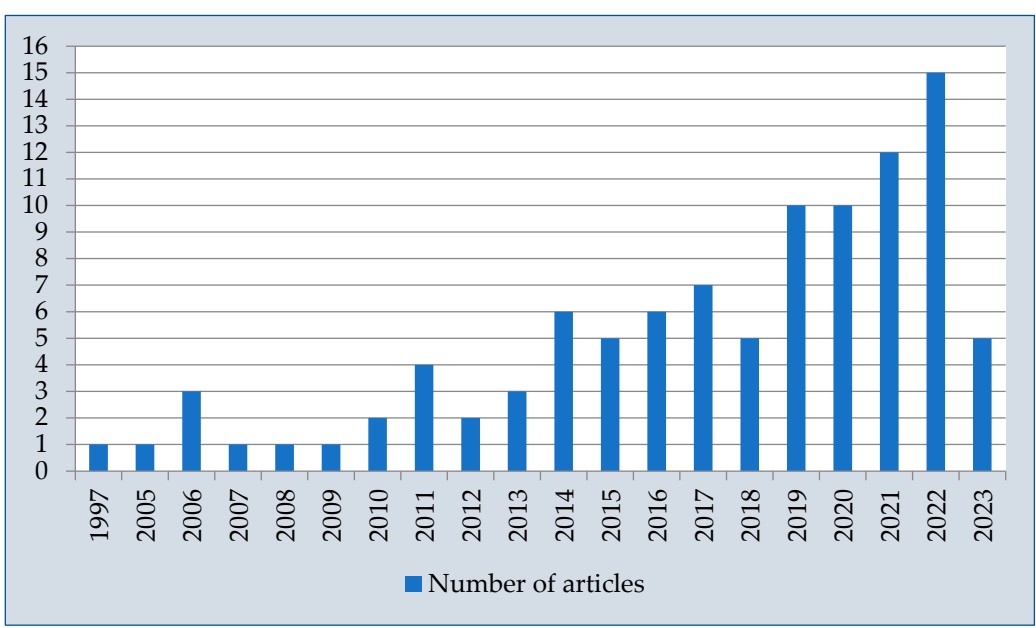

**Figure 1.** Number of studies selected for evaluation by year of publication in all scientific research platforms consulted. Source: the authors (2024).

Results of the Systematic Review

Table 1 indicates how many articles were accepted and rejected at each level.

There has been a significant rise in the number of papers that research HTC in cities with high vertical expansion in the last decade, particularly after 2013. More than ten publications were published in the years 2020, 2021, and 2023, indicating that there is a stronger focus on HTC in urban contexts.

*3.2. Data Meta-Analysis*

The first feature aimed to analyze the origin of the works and the locations where the studies were conducted; hence, Figure 2 depicts the geographic distribution across 33 nations.

The geographical distribution of the works selected for reading and evaluation, as shown in the map, indicates a predominance of HTC investigations at the urban level in the Northern Hemisphere in 28 countries with a total of twenty-three articles, while Brazil, Australia, Chile, and Indonesia (since the study area is Jakarta, which is located in the Southern Hemisphere) have only nine published articles.

China had the greatest number of studies (twenty-eight articles); followed by the United States (eight); Algeria, Germany, and Brazil (four from each); Australia, Bangladesh, Egypt, Greece, India, Iran, and Singapore (three articles each); Canada, Israel, Japan, Morocco, Holland, and Singapore (two articles each); and Saudi Arabia, Chile, Cuba, Ethiopia, France, Hungary, Ireland, Indonesia, England, Israel, Malaysia, Poland, Sri Lanka, Sweden, Taiwan, Turkey, and Ethiopia (one article each).

The location element of urban regions was found in the most diverse cities in which the surveys were created; thus, it was classed in cities situated in locations distant from the coasts and coastal cities. Of the 100 articles reviewed, 68 were in urban regions situated in the nations' more inland cities, while 32 were in cities near coastal metropolitan areas. Figure 3 depicts a graph of the urban areas' proportions of continental and coastal locations.

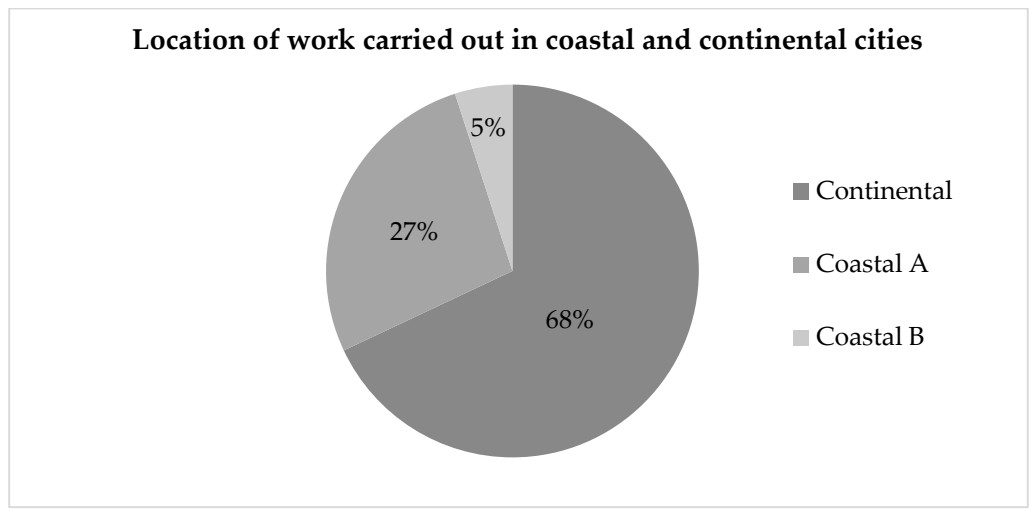

**Figure 2.** Map of the global distribution of countries researching human thermal comfort in urban environments. Source: the authors (2024).

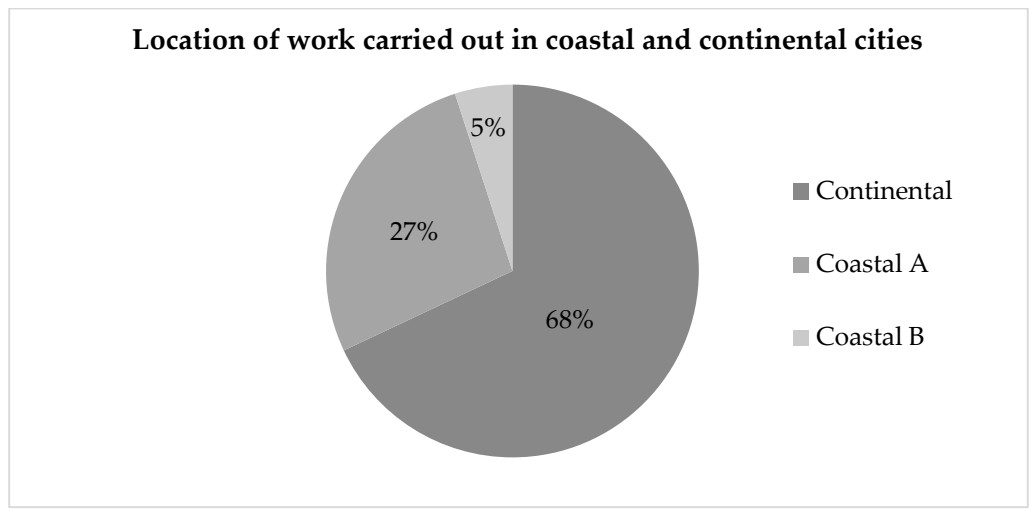

**Figure 3.** Number of surveys carried out in coastal and continental cities. Source: the authors (2024).

The majority of the studies were conducted in metropolitan areas positioned further from the coast. Two scenarios were chosen for the coastal areas: Coastal Cities A and Coastal Cities B. The distinction was important since some works, even in coastal areas, took the closeness to the water into account while others did not. Coastal A: adjacent to

significant bodies of water, although the presence of the sea was not taken into account in the assessment of thermal comfort in the urban area. Concerning the Coastal Cities B demarcation, they were works that had some link with the reality of marine proximity, although most of the time, they created subtle correlations without going into detail. Figure 4 presents the Köppen Climate Classification raised according to each study area in each research.

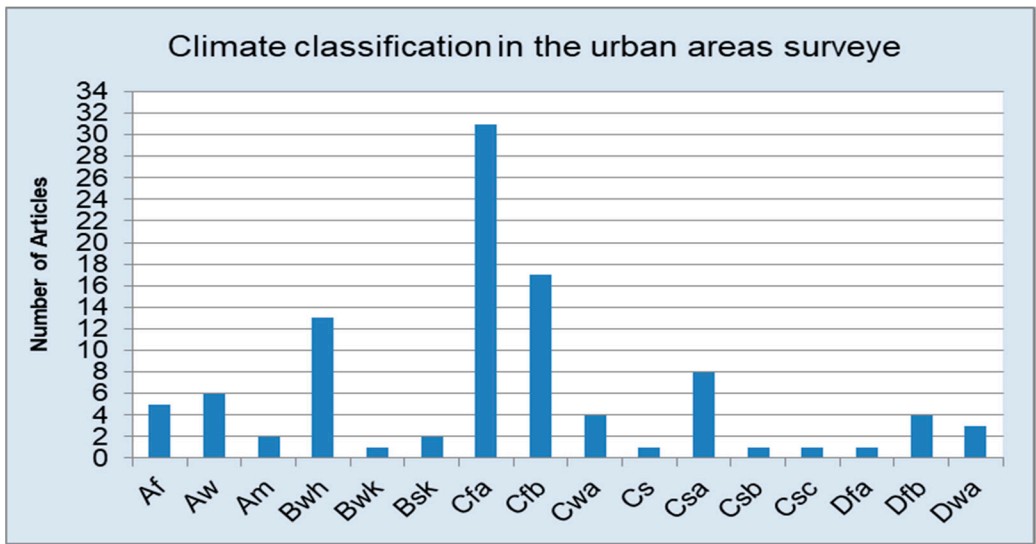

**Figure 4.** Climate Classification of the areas of the cities addressed in the systematic review. Source: the authors (2024).

To characterize the research areas, the Köppen Climate Classification was employed, which typically represents the climatic environment in which the urban area studied is located. Cfa and Cfb were the most often utilized climatic categories, appearing 31 and 17 times, respectively. These were followed by Bwh (thirteen times), Csa (eight times), Aw (six), Af (five times), Cwa and Dfb (four times), Dwa (three times), Bsk and Am (two times), and Bwk, Cs, Dfa, Csb, and Csc (just once). Another key component in HTC study was the season of the year in which the surveys were conducted (Table 2), some of which were hybrid and others in only one season.

**Table 2.** Set of seasons covered in the surveys.

| Seasons | Total Searches | % Occurrence |
|---|---|---|
| Summer | 65 | 65.0 |
| Winter | 2 | 2.0 |
| Autumn | 1 | 1.0 |
| Spring | 1 | 1.0 |
| Summer/Winter | 20 | 20.0 |
| Summer/Winter/Autumn | 2 | 2.0 |
| Summer/Spring | 1 | 1.0 |
| All Seasons | 6 | 6.0 |
| Undefined | 2 | 2.0 |
| Total | 100 | 100.0 |

Summer and winter seasons had the highest amount of study publications, with 70.29% (ninety-seven articles) utilizing summer and 21.73% (thirty articles) using winter, 3.62% (five articles) using spring, and 4.34% (six articles) using autumn. It should be emphasized that only six polls (6.19%) of the total were conducted during the year. Figure 4 depicts the Köppen Climatic Classifications for each survey's second research region. The

graph in Figure 5 depicts the daily duration of data collection used to calculate the thermal comfort indices.

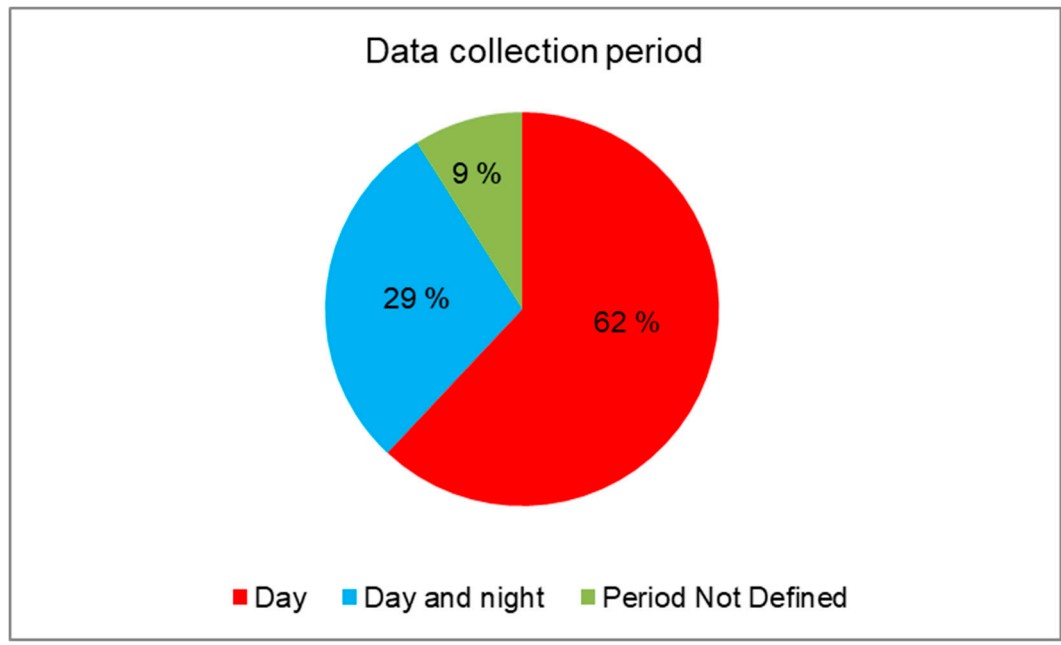

**Figure 5.** Percentage of daily periods most addressed in research on human thermal comfort in urban areas. Source: the authors (2024).

The majority of the works collected and examined data during the day, accounting for (91.0%) of the surveys, with 29.0% of the surveys conducted during both the day and night time. Another 9.0% did not specify the time frame for the study. Figure 6 shows a table with the temperature indices utilized in the various studies.

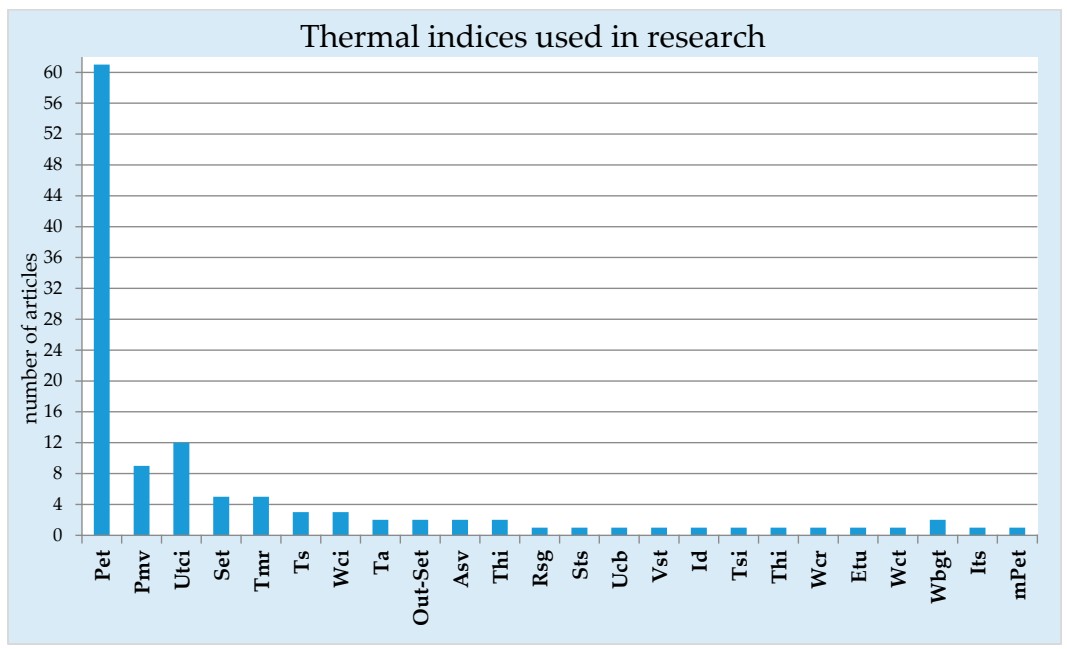

**Figure 6.** HTC indexes most used in research in open areas. Source: the authors (2024).

The indices most used to measure HTC in open environments in highly verticalized cities in the works were Pet (sixty-one); Utci (twelve); Pmv (nine); Set (five); Trm (five); Ts

(three); Wci (three); Ta, Out-Set, Asv, and Asv (two times each); Rsg, Ucb, Vst, Id, Tsi, Thi, Wcr, Etu, Wct, and mPet (used once each index). It is noteworthy that some articles used only the Trm to measure conditions for thermal comfort.

The definitions of application regions are highly significant in open environments and studies on thermal comfort in urban contexts, both with numerical models and in research that gathers interviews. Figure 7 depicts the characteristics used to determine the application zones in urban contexts. Verticalized landscapes with urban canyons were characterized as urban regions.

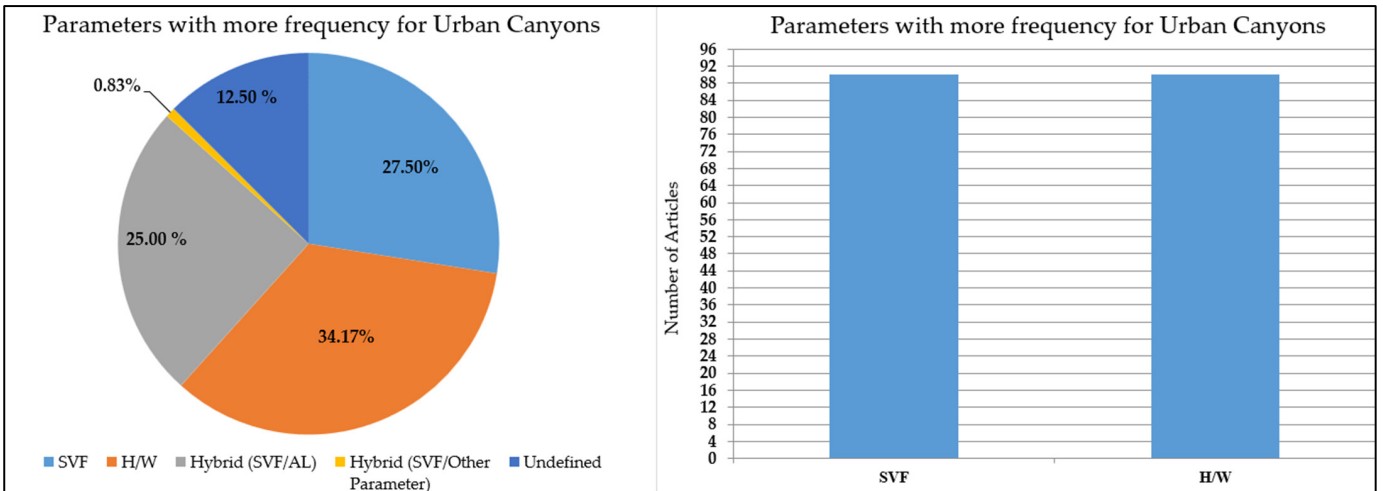

**Figure 7.** Parameters used to define canyons in urban areas. Source: the authors (2023).

The H/W ratio was used ninety times to define and characterize the study areas, eighty times with the SVF, nineteen times with a hybrid strategy utilizing H/W with SVF, and just one time with a hybrid between SVF and another parameter. Figure 8 demonstrates how street orientation was used to describe the urban study environment, whereas Figure 9 depicts how meteorological data were gathered during surveys.

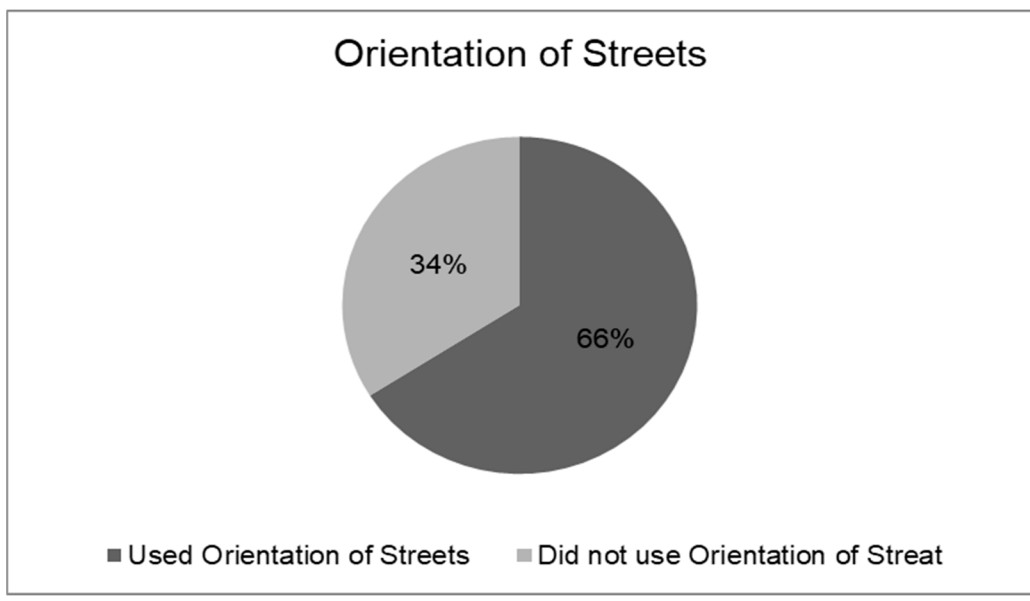

**Figure 8.** Percentage of street orientation as an urban research environment criterion. Source: the authors (2024).

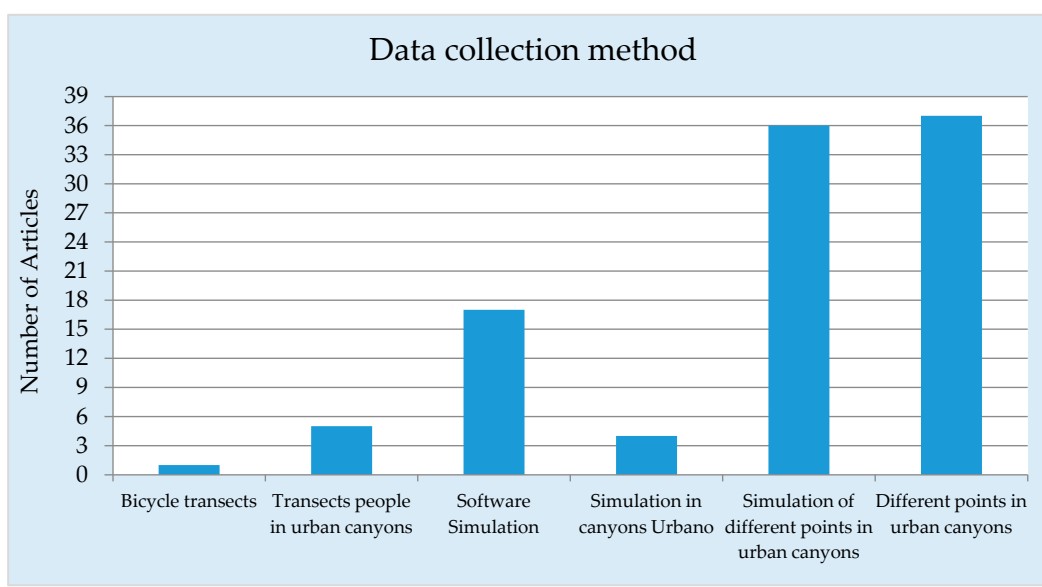

**Figure 9.** Meteorological data collection methods. Source: the authors (2024).

According to Figure 8, a total of 66.0% (*n* = 66) studies employed street orientation as one of the methods to define the urban settings studied, with the change in orientation also influencing air temperature and human thermal comfort. Another 34.0% (*n* = 34) did not believe that the direction of the urban roadway impacted thermal comfort at the pedestrian level.

Data were simulated at different positions in canyons in 36 works, implying that data were acquired with meteorological devices from more than one site in the urban structure. The software simulations at various sites in canyons were carried out in recognized canyons using secondary data acquired from local weather stations. Four works included simulations of canyon building in software, three works had mobile transects of individuals walking in canyons, and one piece included a bicycle transect. Table 3 describes the various data-gathering methods. The equipment used to acquire meteorological data is depicted in Figure 10.

**Table 3.** Description of ways of collecting meteorological data.

| Types of Simulations | Description |
|---|---|
| Software simulation | Software models were used, in which the settings of a canyon were defined as defined parameters, with settings defined by the researcher. such as height, shape, orientation, construction form. |
| Canyon simulation | Software models were used in known canyons. Calibrated the parameters of these canyons. |
| Simulation of different points in canyons | Simulation was performed in different canyons and different points for modeling the simulation of human thermal comfort through software with defined parameters. |
| Different points in canyons | Measurements were carried out in existing canyons. |

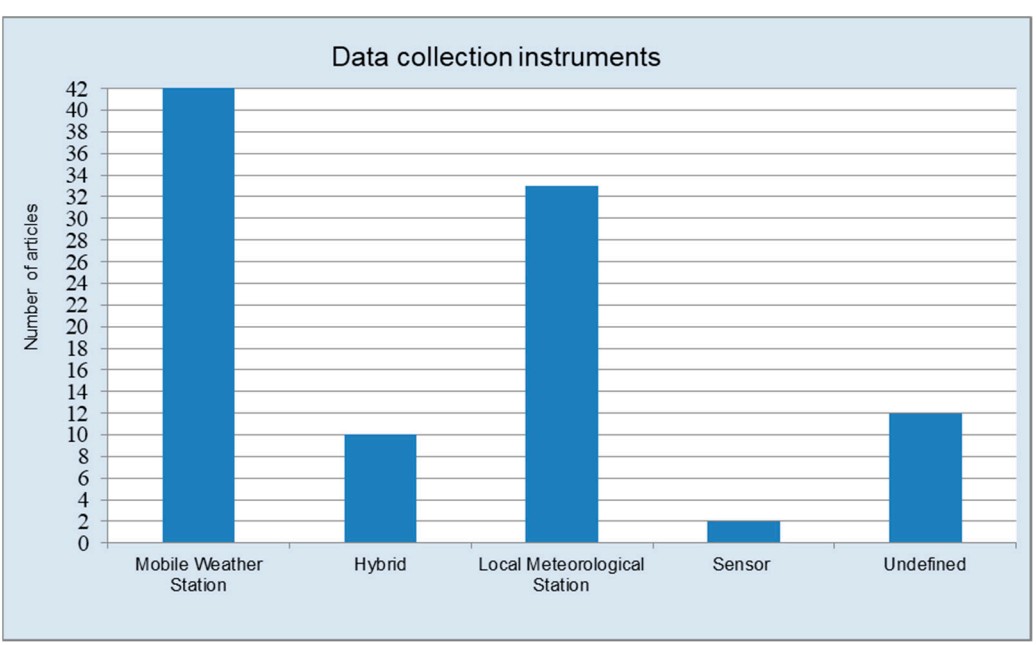

**Figure 10.** Data collection instruments. Source: the authors (2024).

Data were collected with mobile meteorological stations added to the research environment 43 times, 33 times were from fixed local meteorological stations, primarily from airports, and 10 times were hybrid (with local collection and use of data from city stations). The form of data collection was not specified in the other 12 articles. Figure 11 depicts the software used as models to determine HTC indices.

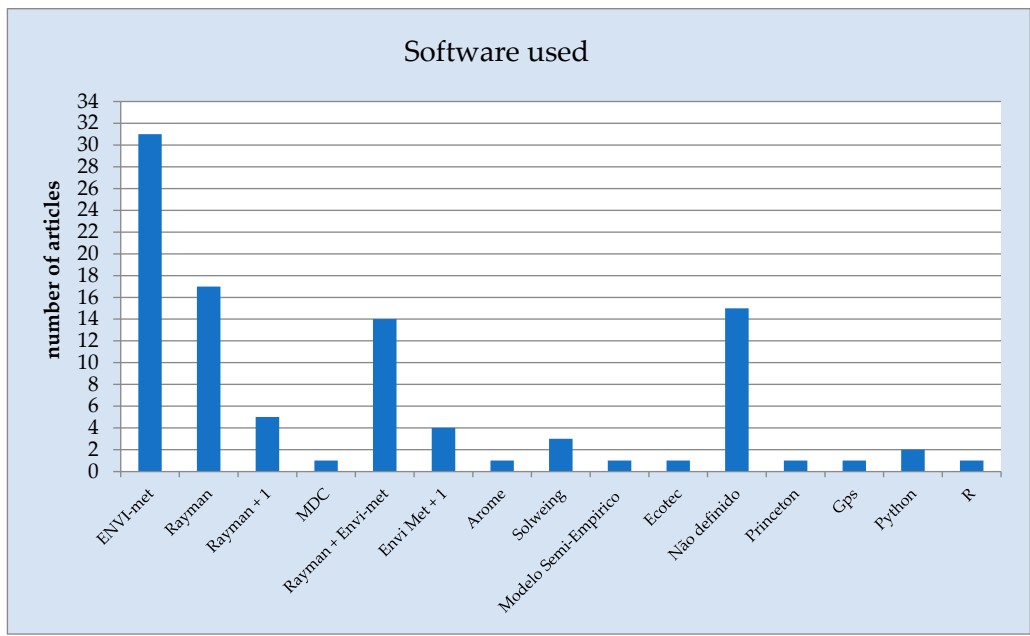

**Figure 11.** Software used in surveys to determine human thermal comfort indices. Source: the authors (2024).

Envi-Met, which featured in 49 publications, and RayMan, which appeared in 36 articles, were the most commonly used software for data tabulation and computation of thermal comfort indices. The models were not recognized in 12 works, and the MDC, Arome, Semi-Empirico Model, Ecotec, Princeton, Gps, BioKlima, JMP15, Flair, R, Python, and ANSYS/Fluent® were only utilized once. Table 4 shows some of the research findings that

only employed the H/W aspect and/or street orientation, and Table 5 resume the meaning of some acronyms used during the review.

**Table 4.** Some investigations that exploited the H/W aspect and/or street orientation yielded research findings found in systematic review.

| Types of Simulations | City/Country | Description |
|---|---|---|
| Cárdenas-Jirón, L. A., Graw, K., Gangwisch, M., and Matzarakis, A. (2023). [1] | Santiago Chile | The current investigation included two interconnected methods: measurement and simulation. The street orientations were EW, NS, NE-SW, and NW-SE, with H/W ratios of 2.5, 1.5, and 0.5. HTC was calculated using the Physiologically Equivalent Temperature (Pet), Modified Physiologically Equivalent Temperature (mPET), and Universal Thermal Climate Index (Utci). Two urban development proposals were evaluated. The two urban development proposals analyzed revealed equivalent patterns of external thermal conditions, with the exception of H/W = 2.5 in Scheme D. Sun exposure is stronger in the winter at a low H/W ratio, but larger quantities reduce stress. For canyons with more than six stories, the NS street axis is favored over any other orientation, while the second-best option is NW-SE, which is also accessible for neighborhood streets with a maximum of six stories. It is appropriate for canyons with more than ten floors and a street width of 20 m on NE-SW-oriented roadways. |
| Wai, K. M., Yuan, C., Lai, A., and Peter, K. N. (2020). [5] | Hong Kong China | H/W—Recommendations for increasing porosity in tropical and subtropical climates to minimize heat stress estimated by the PET index at pedestrian level. |
| Ali-Toudert, F., Djenane, M., Bensalem, R., and Mayer, H. (2005). [32] | Beni-Isguen Algeria | The Petindex was employed in a study conducted on summer days. Streets with varying H/W dimensions were assessed. On warmer days, streets with more sky coverage produced lower values. |
| Emmanuel, R.; Johansson, E (2006). [33] | Colombo Sri Lanka | On warmer days, the authors used H/W with an examination of the Pet index. More pleasant circumstances were observed in narrow estuaries with towering structures to moderate the heat; because of their proximity to the sea, the sea breeze had a good effect. More compact urban forms with deeper street canyons gave pedestrians shade; however, the authors advised caution because night-time cooling and natural ventilation tend to diminish with increased H/W in residential areas. Deeper canyons have less dispersion of pollutants than shallower canyons, which is a drawback. |
| Ali-Toudert, F., and Mayer, H. (2006). [34] | Ghardaia Algeria | The H/W ratio was employed in the research (H/W = 0.5, 1, 2, and 4) as well as street orientation (EW, NS, NE-SW, and NW-S). Findings: The thermal environment is quite demanding and almost fully independent of orientation for broad streets (H/W = 0.5). In the case of EW orientation, it is significantly more stressful. The combination of NS orientation and a high aspect ratio equal to or more than H/W = 2 results in a much-improved thermal environment with lower Pet maxima and markedly shorter times of high stress. Because wall shading is more effective in these situations, NE-SW or NW-SE street orientations give better comfort conditions for the same aspect ratio H/W = 2. |
| Ma, X., Fukuda, H., Zhou, D., Gao, W., and Wang, M. (2019). [35] | Tai Zhou China | H/W and SVF were used. A thermal calendar for tourist visitation is presented, which demonstrates that the entire region is not pleasant for visiting from 8:00 a.m. to 6:00 p.m. in summer. The results show that a deeper canyon (increasing building height: ratio constituted of a high Height × Width ratio (H/W) and a reduced Sky View Factor (SVF)) correlates with a lower level of Pet throughout the day. In severe heat, the combination of morphological tactics and increased plant cover results in more visits in the mornings (between 8:00 and 11:00 a.m.) and after 6:00 p.m. |

**Table 4.** *Cont.*

| Types of Simulations | City/Country | Description |
|---|---|---|
| Jamei, E., and Rajagopalan, P. (2017). [36] | Melbourne Australia | SVF and H/W were used. The aim of the "Melbourne Plan" was to increase human thermal comfort. In deeper canyons with higher aspect ratios and lower Sky View Factors, in the future scenario, the mean radiant temperature was 42.0–64.0 °C and in the current scenario, the mean radiant temperature was (49.0–64.0 °C) 0–60.0 °C, which were shown to contribute to a lower level of mean radiant temperatures. The "Increased building height" scenario improved the Physiological Equivalent Temperature (Pet) by 1.0–4.0 °C. The study also found that changing the ratio of H/W was the most effective technique for reducing Tmrt and Pet during the day. |
| Chatzidimitriou, A., and Yannas, S. (2017). [37] | Thessaloniki Greece | H/W and street orientation were used. Findings: Because their axis is diagonal to the prevailing winds, NS and EW canyons with high aspect ratios are preferred for combining summer sun protection with winter wind protection. If there is an efficient presence of vegetation, aspect ratios between 1.0 and 2.0 appear to be the most advantageous for thermal comfort in NW-SE and NE-SW canyons. |
| Rodríguez-Algeciras, J., Tablada, A., and Matzarakis, A. (2018). [38] | Camagüey Cuba | H/W and street direction. Principal findings: The street orientations with the least amount of heat stress hours have high facade profiles—NS, NE-SW, EW, and SE-NW. Thermal stress is greatest in EW canyons, with high Pet values near 36.0 °C. |
| Bochenek, A., and Klemm, K. (2020). [39] | Lotz Poland | Set, Pmv, Pet, and Utci were all evaluated. The NS and EW canyons were mostly characterized by "cold" and "slightly cool" temperatures. |
| Gaber, N., Ibrahim, A., Rashad, A. B., Wahba, E., El-Sayad, Z., and Bakr, A. F. (2020). [40] | Alexandria Egypt | Based on a case study in a dense historic urban neighborhood, the paper reports on measurements and simulation findings in a street canyon aligned perpendicular to the prevailing wind. It is a coastal city; however, it has nothing to do with the shore. |
| Krüger, E. L., Minella, F. O., and Rasia, F. (2011). [41] | Curitiba Brazil | The influence of urban geometry was evaluated using the SVF to describe the urban environment and the Pet index. In the SVF analysis, it was discovered that on days with higher temperatures, sites with less blockage of the sky, that is, with a higher SVF value, lead to increased heat pain. Wind speed analysis showed that in subtropical locations like the research area, excessive wind speeds during winter might induce thermal discomfort for walkers. |
| Ma, X., Fukuda, H., Zhou, D., and Wang, M. (2019). [42] | Foshan China | On the warmest day of the year, they measured travelers' thermal sensations in the microclimate of the commercially vital pedestrian zone. Seven separate evaluation points were used. During the day, from 10 a.m. to 7 p.m., none of the selected sites were inside the Pet index comfort zone; moreover, in the early morning (8 a.m. to 10 a.m.), all points were pleasant, with the exception of the point with the most open sun access. According to the authors, some ideas for managers and designers based on prior studies that might increase thermal comfort for visitors in connection to open areas are as follows: (A) Increase the average height of buildings to provide greater shade and reduce radiation for tourists. (B) Increase the rate of tree and grass cover to increase cooling and minimize thermal stress. (C) Reduce the rate of paved ground covering to alleviate thermal stress. |

**Table 4.** *Cont.*

| Types of Simulations | City/Country | Description |
|---|---|---|
| Kakon, A. N., Nobuo, M., Kojima, S., and Yoko, T. (2010). [43] | Dhaka Bangladesh | The Temperature Humidity Index (THI) was used to investigate the influence of high buildings on outdoor thermal comfort during the day in summer. The authors utilized an existent urban canyon as well as a canyon model with increased building height (H/W increased and SVF lowered). Because the air temperature in the canyon reduced to some amount as the building's height increased, the Thi became more pleasant as the building's height increased. The temperature lowered and wind speed increased (H/W increased and SVF decreased) at certain hours of the day. The authors note that increasing building height can provide better HTC conditions up to a limit, depending on if there are thermal challenge circumstances to do with urbanization, particularly in densely populated places. |
| Lee, H., Mayer, H., and Kuttler, W. (2020). [44] | Freiburg Germany | The Pet index was utilized to simulate pedestrian-focused human thermal comfort scenarios on the tree-lined sidewalks of a shallow and deep EW roadway canyon. The gain in thermal comfort was larger in areas with smaller tree spacing (better canopy coverage). The shallow street canyon (H/W = 0.5) had greater HTC mitigation than the deep one (H/W = 2.0). |
| Abdelhafez, M. H. H., Altaf, F., Alshenaifi, M., Hamdy, O., and Ragab, A. (2022). [45] | Alexandria and Aswan Egypt | Aspect ratios (H/W) and street canyon orientations, as well as Physiological Equivalent Temperature (Pet), were determined in Alexandria and Aswan, Egypt. The ratios H/W = 2.5 and H/W = 2 in all indicated street canyon orientations in both cities can provide the highest degrees of thermal comfort. |
| Abd Elraouf, R., Elmokadem, A., Megahed, N., Eleinen, O. A., and Eltarabily, S. (2022). [46] | Harbin China | Three typical communities were explored, each with a distinct urban density and traffic layout. Three areas were chosen to symbolize the core area's shared roles and layout: the historic low-rise commercial pedestrian strip (the modern high-rise shopping center, and a medium-sized residential neighborhood). The results showed that the higher the H/W, the greater the comfort level (models with H/W = 2.5 are superior to those with H/W = 1.5 and 1). For street orientations that give shade and the lowest Tmrt, as well as the direction of prevailing breezes, the comfort level can be increased (NS and NW-SE are the most favored street orientations, whereas EW is a poorer orientation than NE-SW). |
| Abdollahzadeh, N., and Biloria, N. (2021). [47] | Liverpool Australia | The purpose of this research was to assess the thermal efficiency of roadways in residential neighborhoods in a subtropical environment in order to increase the Pet index. Street orientation (NS, EW, NE-SW, SE-NW), aspect ratio (0.5, 1, 1.5, 2), building type, and surface coverage were all simulated using computer techniques. The results show that street canyon direction (46.42%), followed by percentage (30.59%), is the most influential component. |
| Abdallah, A. S. H., and Mahmoud, R. M. A. (2022). [48] | New Assiut Egypt | External characteristics in various canyon proportions ranging from 0.24 to 0.6 H/W, as well as modeling of covering with grass, trees, and semi-shade (50%). Outdoor areas in deep canyons accomplish a significant Pet reduction with an H/W ratio of 0.6 compared to shallow canyons with an H/W ratio of 0.24. The three hybrid scenarios that involve the addition of grass, trees, and semi-shade might lower the temperature of the deep canyon by 19.10 °C, 15.0 °C, and 13.6°C, respectively. With an H/W ratio of 0.24, increasing trees or semi-shading by 50% might lower Pet by 17.1 °C and 17.5 °C, respectively. |

| Types of Simulations | City/Country | Description |
| --- | --- | --- |
| Kim, Y. J., and Brown, R. D. (2021). [49] | New York, NY, USA | For pedestrian transect measurements with urban morphology employing H/W and SVF, the human body in Comfort Formula (COMFA index) was employed. The majority of the most thermally unpleasant locations were spatially paired with roadway segments with high SVFs, low H/W ratios, less greenery, and low-density blocks. Thermal stress was quite high on streets with high SVFs, low H/W ratios, and less vegetation with low-density blocks. Terrestrial radiation from walls and the ground surface was the main contribution to thermal loads at the pedestrian level in a deep canyon. |
| Li, Z., Zhang, H., Juan, Y. H., Wen, C. Y., and Yang, A. S. (2022). [50] | Hong Kong China | The effects of horizontal and vertical setbacks on external thermal comfort and air quality were studied concurrently in an urban canyon of low-rise (H/W = 1) and tall (H/W = 2) buildings. The main results were that horizontal setback improves average wind speed at pedestrian level to leeward in the low street canyon (H/W = 1). Furthermore, the average concentration of pollutants on both sides (windward and leeward) at pedestrian level can be lowered by up to 61%. In order to create better outdoor ecosystems, buildings with vertical setbacks are better adapted to canyons. |
| Vassiliades, C., Savvides, A. and Buonomano, A. (2022). [51] | Naples Italy Thessaloniki Greece | As a consequence of the integration of active solar energy systems on existing facades, HTC with Pet in public spaces is being assessed in two coastal towns, Naples and Thessaloniki. The areas are classified using H/W and the direction of the street facade. The effect findings on thermal comfort were better in both cities around the spring equinox. In the summer and fall, Naples has higher heat pain, but Thessaloniki has more vertical shade systems. In the winter, Naples offers better comfort conditions, but Thessaloniki has dismal results. The North–South Street axis is the finest category for both cities. |
| Acero, J. A., Koh, E. J., Ruefenacht, L. A., and Norford, L. K. (2021). [52] | Singapore Singapore | There were 21 scenarios studied, with four H/W ratios (1.5, 2.5, 3, and 3.5) and four distinct street axis orientations (NS, N E-SW, EW, NW-SE). The greatest outcomes for HTC were for H/W ratios between (2.5–3) and on streets with an NS direction, whereas streets with NE-SW orientation caused the most discomfort. |
| Al Haddid, H., and Al-Obaidi, K. M. (2022). [53] | Cardiff and Bristol England | The research concentrated on three unique H/W canyons in Cardiff and Bristol: Deep, Shallow, and Even. Summer Pet values are lower in the specified H/W and SVF settings, according to the results. In the winter, there was an inverse relationship between H/W and SVF, indicating considerable cold stress. |
| Mahmoud, H., Ghanem, H., and Sodoudi, S. (2021). [54] | Aswan Egypt | The Pet index was developed to assess thermal comfort in open areas in five metropolitan shapes and diverse geometric circumstances. H/W aspect ratios of 1, 2, and 4 were given to both NS and EW street orientations, with SVF ranging from 0.05 to 0.26. Thermal comfort was addressed through the development of strategies. The findings show that HTC mitigation techniques work at the pedestrian level in all circumstances. |
| Abreu-Harbich, L. V., Labaki, L. C., and Sampaio, V.H.P., Labaki, L, C., Matzarakis, A. (2014). [55] | Campinas Brazil | In a typical summer scenario, pedestrians passing through two urban canyons were questioned. The preference of pedestrians in this situation was connected to strolling through shady settings within the urban canyon. In hot weather, wind speed and the quantity of covered space were linked to higher HTC. |
| Boumaraf, H., and Amireche, L. (2021). [56] | Biskra Algeria | The behavior of individuals in diverse metropolitan contexts in summer and winter was studied using interviews and filming. The study's key conclusion was that pedestrians spent less time in open surroundings when they were most uncomfortable, both in the cold and in the heat. |

Source: The authors (2024).

**Table 5.** Presents the meaning of some acronyms used during the review.

| Acronyms | Meaning | References Based on Some of the Articles Evaluated in the Systematic Review |
|---|---|---|
| Asv | Actual Sensation Vote | Lamarca, C., Qüense, J., and Henríquez, C. (2018) [57] |
| Etu | Universal effective temperature | Watanabe, S., Nagano, K., Ishii, J., and Horikoshi, T. (2014) [58] |
| H/W | Height × Width Ratio | Jamei, E., and Rajagopalan, P. (2017) [36]; Johansson. (2006) [59]; Paolini, R., Mainini, A. G., Poli, T., and Vercesi, L. (2014) [60]; Sun, C., Lian, W., Liu, L., Dong, Q., and Han, Y. (2022) [61]; Athamena, K. (2022) [62] |
| Id | Discomfort Index | Din, M. F. M., Lee, Y. Y., Ponraj, M., Ossen, D. R., Iwao, K., and Chelliapan, S. (2014) [63] |
| Its | Heat stress index | Gadish, I, Saaroni, H., and Pearlmutter, D. (2023) [64] |
| Lcz | Local Climate zone | Lau, K. K. L., Chung, S. C., and Ren, C. (2019) [65]; Gadish, I, Saaroni, H., & and Pearlmutter, D. (2023) [64]; Yan, H., Wu, F., Nan, X., Han, Q., Shao, F., and Bao, Z. (2022) [66]; |
| mPet - | Modified Physiologically equivalent temperature | Cárdenas-Jirón, L. A., Graw, K., Gangwisch, M., and Matzarakis, A. (2023) [1] |
| Pet | Physiologically equivalent temperature | Chatzidimitriou, A., and Yannas, S. (2017) [37]; Krüger, E. L., and Rossi, F. A. (2011) [41]; Deevi, B., &and Chundeli, F. A. (2020) [67]; Kim, Y. J., & and Brown, R. D. (2021) [49]; Abreu-Harbich, L. V., Labaki, L. C., and Sampaio, V.H.P., Labaki, L, C., Matzarakis, A.(2014) [55] |
| Out-Set | Standard Effective Temperature | Watanabe, S., Nagano, K., Ishii, J., and Horikoshi, T. (2014) [58] |
| Pmv | Predicted Mean Vote | Jihad, A. S., and Tahiri, M. (2016) [68]; Gaber, N., Ibrahim, A., Rashad, A. B., Wahba, E., El-Sayad, Z., and Bakr, A. F. (2020) [40]; Limona, S.S, Al-hagla, K. S., and El-sayad, Z. T. (2019) [69] |
| Ptci | Perceptual thermal comfort index | Lamarca, C., Qüense, J., and Henríquez, C. (2018) [57] |
| Rsg | Global Solar Radiation | Hwang, R. L., Lin, T. P., and Matzarakis, A. (2011) [70]; Yin, S., Lang, W., and Xiao, Y. (2019) [71]; Chen, L., Yu, B., Yang, F., and Mayer, H. (2016) [72] |
| Set | Standard Effective Temperature | Bochenek, A., and Klemm, K. (2020) [39]; Rosheidat, A., Hoffman, D., and Bryan, H. (2008) [73]; Ali-Toudert, F., Djenane, M., Bensalem, R., Mayer, H. (2005) [32] |
| SVF | Sky View Factor | Ma, X., Fukuda, H., Zhou, D., Gao, W., and Wang, M. (2019) [42]; Jamei, E., and Rajagopalan, P. (2017) [36]; Bochenek, A., and Klemm, K. (2020) [39]; Rosheidat, A., Hoffman, D., and Bryan, H. (2008) [73] |
| Ta | Apparent temperature | Pioppi, B., Pigliautile, I., and Pisello, A. L. (2020) [74] |
| Thi | Temperature-humidity Index | Mirzaei, P. A., and Haghighat, F. (2012) [75]; Kakon, A. N., Nobuo, M., Kojima, S., and Yoko, T [43] |
| Trm | Mean radiant temperature | Boumaraf, H., and Amireche, L. (2021) [56]; Wai, K. M., Yuan, C., Lai, A., and Pe-ter, K. N. (2020) [5] |
| Ucb | Berkeley thermal comfort model | Lee, H., Mayer, H., and Kuttler, W. (2020) [44]; Huang, K. T., and Li, Y. J. (2017) [9] |
| Utci | Universal Thermal Climate Index | Paolini, R., Mainini, A. G., Poli, T., and Vercesi, L. (2014) [60]; Deevi, B., and Chundeli, F. A. (2020) [67]; Latini, G., Grifoni, R. C., and Tascini, S. (2010) [76]; Krüger, E. (2017) [77]; Croce, S., D'Agnolo, E., Caini, M., and Paparella, R. (2021) [78] |
| Vcg | Vote of general comfort | Yao, J., Yang, F., Zhuang, Z., Shao, Y., and Yuan, P. F. (2018) [79] |
| Vst | Thermal sensation votes | Yao, J., Yang, F., Zhuang, Z., Shao, Y., and Yuan, P. F. (2018) [79] |
| Wbgt | Wet Bulb Globe Temperature | Deng, X., Cao, Q., Wang, L., Wang, W., Wang, S., and Wang, L. (2022) [80] |
| Wci | Wind Chill Index | Mirzaei, P. A., and Haghighat, F. (2012) [75] |
| Wct | Wind chill temperature | Liu, Y., Jin, H., and Xu, X. (2019) [81] |

Source: The authors (2024).

Figure 12 depicts the number of works that employed questionnaire data collecting and the number of participants questioned.

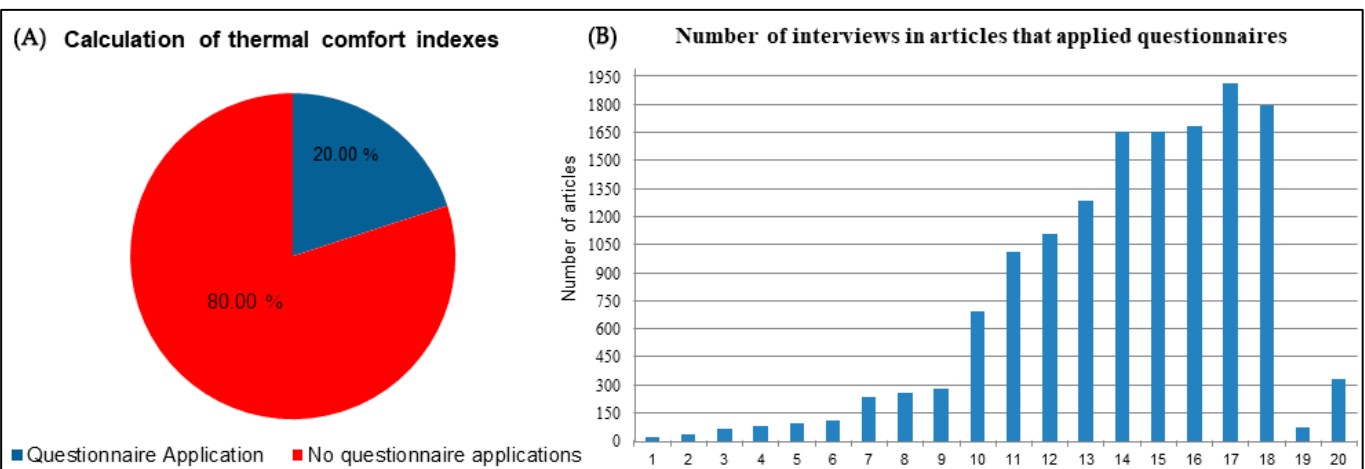

**Figure 12.** (**A**) Percentage of surveys that used thermal preference questionnaires on pedestrians in urban areas and (**B**) number of people interviewed in each article. Source: The authors (2024).

Only 20.0% (20 articles) of the surveys employed interviews with individuals to calibrate the thermal indices used, whereas 80.0% (80 articles) calculated research indices of the population's personal preference, and most simulations were performed with established parameters. The biggest number of interviews gathered among the 20 works was 1917 interviews, while the smallest number was 26 persons. All in all, eleven papers interviewed less than 1000 people, with just eight exceeding that number. Figure 13 depicts a map of the nations where pedestrians were questioned in urban settings.

Only eight nations gathered pedestrian preferences. In the Northern Hemisphere, China has six articles; Bangladesh, India, and Japan each have two; and Algeria, Iran, and Malaysia each have one. Only Brazil published four studies with interviews of people outside in metropolitan settings in the Southern Hemisphere. Pet and Utci were the most often utilized thermal indices in studies on thermal comfort in outdoor urban environments. Figure 14 depicts which heat indices were used during human studies.

Pet and Utci were the most commonly used indices in publications that investigated thermal comfort in outdoor urban environments, with ten and six studies, respectively, using them. Other indices used were Set and TS in two articles, and Out-Set, Pmv, UCb, Vst, Id, Asv, and Etu once each. In six of the publications, more than one indicator was used to measure thermal comfort, whereas in eleven of the articles, only one index was employed. In terms of data-collecting time, 95.0% (nineteen articles) took place during the day, while only 5.0% (one article) conducted interviews that continued till the night.

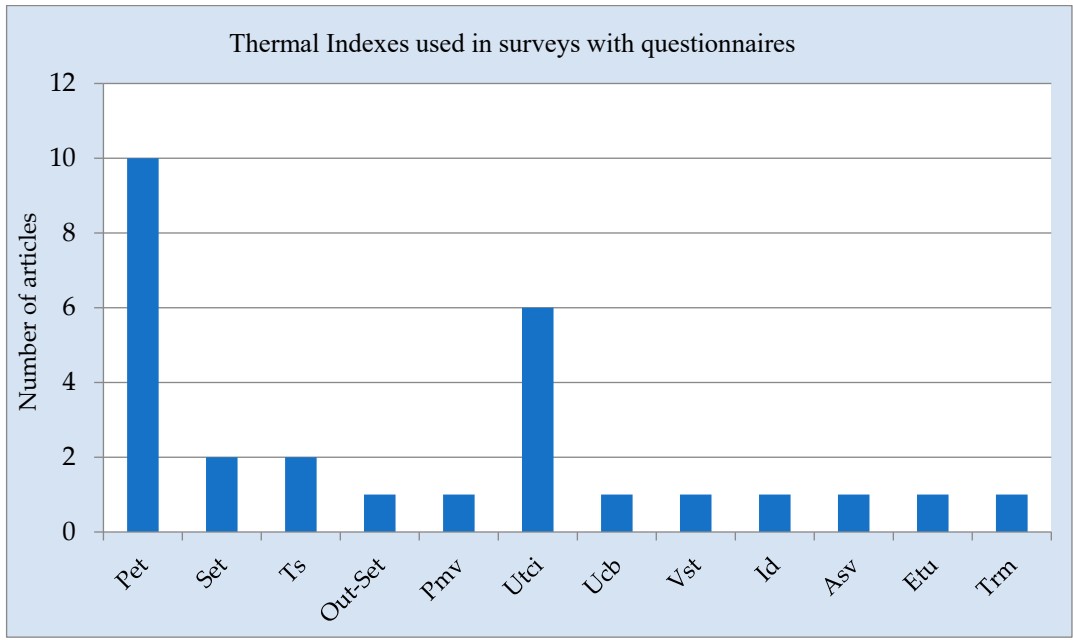

**Figure 13.** Countries that conducted interviews with pedestrians in urban environments. Source: the authors (2024).

**Figure 14.** Thermal indices used in research when they revealed the use of interviews in their procedures. Source: the authors (2024).

## 4. Discussion

The urban environments with the highest concentration of research related to HTC in their open areas were predominantly located in the Northern Hemisphere, as well as in Europe, Asia, and the northern part of the African continent. It is thought that the process of urbanization of cities in the Northern Hemisphere's continental sections has taken longer than that of cities in the Southern Hemisphere's continents. However, according to [82], the Southern Hemisphere is quickly urbanizing, and the global south was responsible for 94% of the rise in the worldwide population between 2010 and 2015, and another especially intriguing statistic is that now, of the 33 global megacities, 27 are in the global south.

There has been a rise in the number of studies on HTC in metropolitan areas in recent years, particularly in the last decade. External areas must be of good quality and livable in order for people to carry out their daily activities in greater comfort [83]. Thus, a concern for cities' and their inhabitants' health arises, represented by HTC in outdoor spaces, which has been the subject of countless studies in recent decades, primarily in cities with a high degree of urbanization and verticalization [11–14,84–86].

The larger the number of studies that cover summer and daylight hours to evaluate HTC in metropolitan settings, the greater the worry about warming and climate change. Notably, a more focused concern about warmer times is justified, according to the Intergovernmental Panel on Climate Change (IPCC) [87] estimates, which show an increase in the world average temperature for the foreseeable future. While climate change affects all human populations, it can have the most dramatic consequences on large cities with higher population densities [87–89].

Urban parks influence exterior thermal comfort in the context of global warming, with the finding that metropolitan areas are warmer than surroundings and rural regions [90], owing mostly to the establishment of urban heat island effects (UHI). In light of climate change, the author suggests that managers must enhance realistic options for cooling urban parks.

Changing the morphology of cities affects environmental variables such as solar radiation, air temperature, humidity, and wind speed. To classify urban environments, [6] developed the Sky View Factor (SVF), and [91] developed the Height Relation x Width (H/W). These strategies are the primary methods for categorizing and defining urban regions in the urban climatic environment, as well as for defining urban canyons. It is worth noting that the SVF and the H/W ratio are both used as categorization processes in the LCZ technique.

The main key concept for determining urban canyons is the ratio calculated by calculating the average height between two buildings, which is given by the ratio between the average height of two buildings (A) divided by the width of the street between these two buildings (L) [41]. Some studies link the HTC to the proportion of canyons, such as [59] in Morocco, which found that deeper canyons provided better thermal conditions in the summer while shallow canyons provided better thermal conditions in the winter. Deep canyons, synonymous with shaded streets, have been shown to improve pedestrian thermal comfort in studies conducted by [32] in Beni-Isguen, Algeria; [33] in Colombo, Sri Lanka; [34] in Ghardaia, Algeria; [92] in Freiburg, Germany; and [93] in Taipei, Taiwan.

The SVF scaled from 0 to 1, with 1 being an open sky setting with no obstructions to shade the surfaces and 0 representing a state with a reduced capacity to view the sky [91,94]. According to Jamei et al. (2016) [95], the SVF within urban buildings is always less than 1 due to the existence of impediments generated by urban geometry.

Thermal indices developed for indoor environments may potentially be used to measure conditions in outdoor areas, with some changes required owing to the higher complexity of extra components. The authors of [96] discovered in a review that only four of the one hundred and sixty-five human thermal indices developed are widely used in research on thermal perception in open environments, namely the Physiological Equivalent Temperature (Pet), Predicted Mean Vote (Pmv), Universal Thermal Climate Index (Utci), and Standard Effective Temperature (Set), due to their applicability [96]. These indices

were the most commonly employed in this systematic review, both in the overall evaluation and in the selection of papers that utilized indices calibrated using personal surveys.

## 5. Conclusions

The urban environments with the highest concentration of research related to HTC in their open areas were predominantly located in the Northern Hemisphere, as well as in Europe, Asia, and the northern part of the African continent, and only the United States and Canada in the northern portion of the American continent, countries considered already developed. It is thought that the process of urbanization of cities in the continental sections of the Northern Hemisphere is older and that is why cities have been consolidated for longer than the urbanization of countries in the Southern Hemisphere, which are still developing. However, according to [82], the Southern Hemisphere is rapidly urbanizing, and the global south was responsible for 94% of the increase in the world population between 2010 and 2015, and another especially intriguing statistic is that now, out of 33 megacities globally, 27 are in the global south. The three basic urban area categorization approaches are the H/W ratio, SVF, and LCZ. All these approaches are useful for categorizing urban areas, particularly those with compact and concentrated buildings, which is a prevalent feature of urbanized cities. The introduction of LCZs, a more recent and increasing categorization approach that includes both H/W and SVF on its platform, stands out. According to [97], it is becoming increasingly significant in urban climate research since it offers a universal categorization method for both natural and urban areas.

According to [98], urbanization originated in the Northern Hemisphere, both in North America and Western Europe, but it now occurs more strongly in emerging nations, which are largely in the Southern Hemisphere. According to the findings of this study, the majority of research on HTC in verticalized urban environments has been conducted in Northern Hemisphere cities, such as cities on the European continent, the Asian continent, China, and the North American continent, the United States. On the other hand, the HTC method is becoming more popular in urban regions than in nations in the Southern Hemisphere, with a focus on Brazil and Chile in Latin America and Australia in Oceania.

**Author Contributions:** Conceptualization, I.T.C., L.W., C.A.W. and J.P.A.G.; methodology, L.W., C.A.W., L.W., A.C.I., A.N.d.S., O.d.F.B. and J.P.A.G.; software, I.T.C., L.W. and C.A.W.; validation, I.T.C., C.A.W., L.W., A.C.I., A.N.d.S., O.d.F.B., J.P.A.G., S.S. and A.M.; formal analysis, A.M. and S.S.; investigation, I.T.C. and C.A.W.; resources, I.T.C. and C.A.W.; data curation, I.T.C., C.A.W. and J.P.A.G.; writing—original draft preparation, I.T.C. and C.A.W.; writing—review and editing, I.T.C., C.A.W., L.W., A.C.I., A.N.d.S., O.d.F.B., J.P.A.G., S.S. and A.M.; visualization, C.A.W., J.P.A.G., S.S. and A.M.; supervision, C.A.W., J.P.A.G., S.S. and A.M.; project administration, C.A.W.; funding acquisition, C.A.W. and S.S. All authors have read and agreed to the published version of the manuscript.

**Funding:** This study was financed in part by the Coordenação de Aperfeiçoamento de Pessoal de Nível Superior-Brasil (CAPES)-Finance Code 001. Conselho Nacional de Desenvolvimento Científico e Tecnológico (CNPq) for proving the Research and Productivity research: grant process number 306505/2020-7.

**Data Availability Statement:** The data presented in this study are available on request from the corresponding author (accurately indicate status).

**Acknowledgments:** We thank the Conselho Nacional de Desenvolvimento Científico e Tecnológico (CNPq) for proving the Research and Productivity research: grant process number 306505/2020-7.

**Conflicts of Interest:** The authors declare no conflicts of interest.

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
