# Peer review of "A Systematic Review on Human Thermal Comfort and Methodologies for Evaluating Urban Morphology in Outdoor Spaces"

_climate, doi:10.3390/cli12030030_

Round 1

Reviewer 1 Report

Comments and Suggestions for Authors

Review of the paper A systematic review on human thermal comfort and methodologies for evaluating urban morphology in outdoor spaces by Iago Turba et al.

 With the growth of cities, microclimate has been changing with the diversity of morphologies in the urban environment, which is more and more important for the human and livings. Therefore, focused on human thermal comfort and methodologies for evaluating urban morphology in outdoor spaces, the manuscript is interesting and has the scientific significance. What’s more, the manuscript contains much sufficient content to support the conclusion. However, the followings must be solved before consideration for publication.

 According to manuscript, all the sources for most the figures are from “The authors (2023)”. What publication does it refer to? If it is only one article, does this manuscript relate to duplicate publications? If it is only one book, the manuscript perhaps should be book review, not one article. If it is a few articles, it is better to be written out respectively. Therefore, I think it is better to give the differences description, or at least note which publication published previously.

 The whole manuscript has poor logicality. At the same time, the 3.2 part concludes most of the results, which makes it quite difficult to read clearly.

 The serial numbers are confusing. For example, there are two 3.1, and the 5 corresponding part is missing.

Comments on the Quality of English Language

Review of the paper A systematic review on human thermal comfort and methodologies for evaluating urban morphology in outdoor spaces by Iago Turba et al.

 With the growth of cities, microclimate has been changing with the diversity of morphologies in the urban environment, which is more and more important for the human and livings. Therefore, focused on human thermal comfort and methodologies for evaluating urban morphology in outdoor spaces, the manuscript is interesting and has the scientific significance. What’s more, the manuscript contains much sufficient content to support the conclusion. However, the followings must be solved before consideration for publication.

 According to manuscript, all the sources for most the figures are from “The authors (2023)”. What publication does it refer to? If it is only one article, does this manuscript relate to duplicate publications? If it is only one book, the manuscript perhaps should be book review, not one article. If it is a few articles, it is better to be written out respectively. Therefore, I think it is better to give the differences description, or at least note which publication published previously.

 The whole manuscript has poor logicality. At the same time, the 3.2 part concludes most of the results, which makes it quite difficult to read clearly.

 The serial numbers are confusing. For example, there are two 3.1, and the 5 corresponding part is missing.

Author Response

Dear reviewer, the answers to your questions and suggestions can be found in the attached text. Thank you very much for your contributions.

Reviewer 2 Report

Comments and Suggestions for Authors

Human thermal comfort and/or discomfort in urban areas is a research topic which is really frequently undertaken. However, the reviewed paper does not provide any added value. This is rather random selection of articles printed after 1997. The geographical distribution of analysed papers is not representative in global research of urban bioclimate. May be this is an effect of narrow list of keywords used. Many interesting papers could be find under keywords of: urban bioclimate, urban heat island or urban morphology measures.

There are also many detail remarks, e.g.

- the introduction is very chaotic; it needs fundamental improvements,

- the consideration of the theory of systematic review is not necessary,

- while in table 2 you consider only 100 papers (table 1 tell about 113 selected papers),

- abbreviations of thermal indices, models and urban morphology measures must be unified in the mapper and explained in special table while some of them are not clear for reader,

- use the same way to write those abbreviation (e.g. PET or Pet) both, in text, tables and figures,

- in table 4 is not  clear what represent cited values of future and present scenario (Melbourne),

- the first sentence of discussion is absolutely not clear (while USA, Europe and Asia are areas of Northern hemisphere).

Author Response

(The authors gave the same response as above.)

Round 2

Reviewer 2 Report

Comments and Suggestions for Authors

In table 5 you must correct explanation to Wci. Must be Wind Chill Index

Author Response

Dear reviewer. Thank you very much for your contributuions. Please, see the answers to your questions attached.
